# The Impact of Psychological Interventions with Elements of Mindfulness (PIM) on Empathy, Well-Being, and Reduction of Burnout in Physicians: A Systematic Review

**DOI:** 10.3390/ijerph182111181

**Published:** 2021-10-25

**Authors:** Sara Tement, Zalika Klemenc Ketiš, Špela Miroševič, Polona Selič-Zupančič

**Affiliations:** 1Department of Family Medicine, Faculty of Medicine, University Maribor, Taborska 8, 2000 Maribor, Slovenia; sara.onuk@gmail.com (S.T.); zalika.klemenc.ketis@gmail.com (Z.K.K.); 2Department of Family Medicine, Faculty of Medicine, University Ljubljana, Poljanski Nasip 58, 1000 Ljubljana, Slovenia; spela.mirosevic@mf.uni-lj.si; 3Community Healthcare Centre Ljubljana, Primary Healthcare Research and Development Institute (IRROZ), Metelkova 9, 1000 Ljubljana, Slovenia; 4Department of Psychology, Faculty of Medicine, University Maribor, Taborska 8, 2000 Maribor, Slovenia

**Keywords:** mindfulness, psychological intervention, physicians, empathy, burnout, well-being, MBSR

## Abstract

**Introduction:** Physician’s burnout has been recognized as an increasing and significant work-related syndrome, described by the combination of emotional exhaustion (EE) and depersonalization (D), together with low personal accomplishment (PA). It has many negative consequences on personal, organizational, and patient care levels. This systematic review aimed to analyze research articles where psychological interventions with elements of mindfulness (PIMs) were used to support physicians in order to reduce burnout and foster empathy and well-being. **Methods:** Systematic searches were conducted in May 2019, within six electronic databases PubMed, EBSCOhost MEDLINE, PsycArticles, Cochrane Library, JSTOR, and Slovenian national library information system. Different combinations of boolean operators were used—mindfulness, empathy, medicine/family medicine/general practice/primary care, burnout, doctors/physicians, intervention, and support group. Additional articles were manually searched from the reference list of the included articles. Studies with other healthcare professionals (not physicians and residents) and/or medical students, and those where PIMs were applied for educational or patient’s treatment purposes were excluded. **Results:** Of 1194 studies identified, 786 screened and 139 assessed for eligibility, there were 18 studies included in this review. Regardless of a specific type of PIMs applied, results, in general, demonstrate a positive impact on empathy, well-being, and reduction in burnout in participating physicians. Compared with other recent systematic reviews, this is unique due to a broader selection of psychological interventions and emphasis on a sustained effect measurement. **Conclusions:** Given the pandemic of COVID-19, it is of utmost importance that this review includes also interventions based on modern information technologies (mobile apps) and can be used as an awareness-raising material for physicians providing information about feasible and easily accessible interventions for effective burnout prevention and/or reduction. Future research should upgrade self-reported data with objective psychological measures and address the question of which intervention offers more benefits to physicians.

## 1. Introduction

In recent years, physician burnout, a result of long-term exposure to emotional and interpersonal stressors at the workplace, has been recognized as an increasing and significant work-related syndrome [1,2]. It was described as the combination of emotional exhaustion (EE) and depersonalization (D), together with low personal accomplishment (PA) [3]. Recently, 61% of European healthcare professionals was reported experiencing work-related stress [4,5]. During the COVID-19 pandemic, the burden of burnout has become even more evident in physicians, who have been continuously facing limited resources, longer shifts, and disruptions to work–life balance, and sleep; all these factors have increased burnout [4,5].

Physician burnout has been widely studied throughout the world [1,6,7]. Many international studies indicated that specialties in the front lines such as primary care are at the highest risk [7,8,9]. Burnout prevalence varies in different studies; Twellaar et al. reported that almost 20% of Dutch family physicians (GPs) are suffering from burnout [10], whereas McCray et al. concluded that up to 60% of physicians experienced burnout at some point in their careers [11]. A study on burnout among 127 Slovenian family medicine trainees showed that 46% of participants reported high EE, 43% high D, and 46% low PA, with more than 18% experiencing burnout in all three dimensions, 28% in two dimensions, 25% in one dimension, and only 29% did not score high for burnout in any dimension at all [12]. These high burnout rates in primary care can be related to insufficient work resources, lower wages, and fewer career development opportunities [8,13].

Burnout has negative consequences on personal, organizational, and patient care levels. Studies report that it is associated with an increased likelihood of a self-perceived medical error [14], suboptimal patient care [15], and lower career satisfaction [7]. At the organizational level, it is often associated with higher rates of absenteeism, attrition, higher staff turnover, and reduced productivity [16].

On the other hand, there has been an increasing number of psychological interventions developed and studied in order to empower professionals coping with increasing work-related stress and burnout [17,18,19]. Recent research suggests that mindfulness training can be very promising in reducing stress and promoting self-care and well-being [20].

### 1.1. Mindfulness and Psychological Interventions with Elements of Mindfulness

Mindfulness is defined as purposefully paying attention to each moment in a non-judgmental way [21] and was first presented by Jon Kabat-Zinn, who introduced mindfulness in a medical setting in the 1970s as an 8-week, group-based program called mindfulness-based stress reduction (MBSR). MBSR was initially offered to patients with chronic and somatic illnesses, such as chronic pain, psoriasis, and anxiety [22,23,24]. Later, research showed that MBSR intervention may be effective for reducing stress and increasing quality of life also in health care professionals [25]. Some recent reviews confirmed that MBSR could benefit physicians in various ways, e.g., decreasing stress levels, burnout, and anxiety and increasing their personal well-being, empathy, and self-compassion, as well as enhancing compassion when relating to others [26,27,28].

### 1.2. Aim and Research Questions

This systematic review aimed to analyze all recently published literature in which psychological interventions with elements of mindfulness (PIMs) were used for physicians, including residents, in order to reduce burnout and foster empathy and well-being. The main advantage of the review is that it offers a thorough insight into these interventions. The topic is of utmost importance, as the COVID-19 pandemic heightened existing challenges for physicians, e.g., increasing workload, which was shown to be directly associated with increased burnout [29].

Another unique characteristic of the review is that in-person and modern information technologies based interventions (mobile apps) were analyzed. Given that studies showed online and other applications/interventions effectively ameliorate burnout and alleviate anxiety and depression in physicians and medical residents, these findings have an additional value in the context of COVID-19 reality. Therefore, burdened physicians are informed about effective and feasible interventions provided to help them improve their well-being. Moreover, online interventions are easily accessible, user friendly, and less time consuming [30].

This review also stresses the importance of the sustained effect of the selected PIMs. Regarding the type of intervention, this review is a unique achievement in comparison with others, since all psychological interventions with elements of mindfulness (PIMs) were included and provided the ground for broader understanding of PIMs, which is an additional value, compared with a sole demonstration of the MBSR effects.

The following questions were answered with this review:
Are PIMs for physicians effective in reducing burnout and increasing empathy and well-being?What is the difference in effect between PIMs regarding their duration, setting (individual/group), and mode of administration (online vs. in person)?


## 2. Methods

This review paper is reported in accordance with the Preferred Reporting Items for Systematic Review and Meta-Analysis (PRISMA) guidelines. A study protocol with a number 150,610 was registered at the International Prospective Register of Systematic Reviews (PROSPERO).

### 2.1. Search Strategy

Systematic searches were conducted from 15 to 31 May 2019, within six electronic databases: PubMed, EBSCOhost MEDLINE, PsycArticles, Cochrane Library, JSTOR, and Cobiss (Slovenian national library information system). Different combinations of Boolean operators were used, such as mindfulness, empathy, medicine/family medicine/general practice/primary care, burnout, doctors/physicians, intervention, and support group. Additional articles were hand-searched from the reference list of the included articles.

The detailed search strategy is outlined in Figure 1 and described as follows:

On PubMed, there were 246 papers in total from various searches: mindfulness “and” intervention “and” doctors (*n* = 213); mindfulness “and” doctors “and” empathy (*n* = 33). Given the search limits, papers published in a 10-year period (2009–2019), written in English, and related to humans were included.

On EBSCOhost, 247 papers were found using the following keywords: doctors “and” mindfulness (*n* = 19); mindfulness “and” doctors “and” empathy (*n* = 228), with search limitation to publications in the last five years.

In PsycArticles, a total of 417 research papers were published in the last 10 years using the following keywords: support group “and” medicine “and” empathy (*n* = 263); mindfulness “and” medicine “and” empathy (*n* = 73); mindfulness “and” family medicine (*n* = 81), which were extracted.

In Cochrane Library, there were 96 systematic reviews found for mindfulness (intervention), while in JSTOR, a total of 154 publications for the last 10 years were identified using the following keywords: mindfulness “and” doctors “and” intervention (*n* = 42); mindfulness “and” empathy “and” doctors (*n* = 21); mindfulness “and” medicine “and” empathy (*n* = 61); mindfulness “and” intervention “and” medicine “and” doctors (*n* = 30); in COBISS, 34 papers were found using different search terms as follows: mindfulness, medicine (*n* = 16); mindfulness, burnout (*n* = 8); mindfulness, empathy (*n* = 10).

### 2.2. Inclusion and Exclusion Criteria

Research studies with (i) quantitative, qualitative, and mixed-method design, (ii) written in English or Slovenian language, (iii) published in last 10 years (or 5 years for EBSCOhost only), and (iv) presenting original research focused on the evaluation of psychological interventions with elements of mindfulness (PIMs), implemented with the intention of burnout reduction and well-being promotion in physicians, including residents, were included in this review.

Studies with (i) psychological interventions used for patients’ treatment or educational purposes and (ii) those focused on other healthcare professionals (not physicians and/or residents) and medical students were also excluded.

### 2.3. Data Extraction

Data extraction was conducted independently by two reviewers (S.T., Š.M.). In the case of disagreement between these reviewers, there was a third reviewer included (P.S.-Z.).

For eligible studies, the following data were extracted: author and year in which the study had been conducted, sample size, topic, study question and study type, experimental design, the medical specialty of participants, intervention characteristics, study methods, instruments, and measures, and main findings.

### 2.4. Quality Assessment

The quality assessment of all included studies was assessed using the Medical Education Research Study Quality Instrument (MERSQI) [31], a 10-item instrument for qualitative studies evaluation, using study design, sampling, data type, the validity of assessments, data analysis, and outcomes as criteria. A maximum of 3 points in each domain with a total of 18 points could have been appointed to a study.

## 3. Results

### 3.1. Search Results and Study Characteristics

Of 1194 articles screened, 18 studies met the inclusion criteria and were analyzed. Though all included studies were interventional in nature, there were differences in experimental design: thirteen studies used a quantitative design, three used a mixed-method design, and two studies applied a qualitative design.

While Appendix A Appendix A (physicians) and Appendix A Appendix A (residents/interns) provide an overview of the characteristics of these studies, i.e., information was gathered about study characteristics—namely, author, year of publication, topic/study question, study design, sample size, population, and country where the study had been conducted, as well as intervention—namely, type (mindfulness-based stress reduction (MBSR), stress management and resilience training (SMART)), duration, follow-up, instruments/measures, and main findings. Table 1 summarizes the main features of the studies.

### 3.2. Assessing Study Quality—MERSQI Scores

The MERSQI scores varied, from a minimum of 5.5 points to a maximum of 14 points, with a mean value of 10.5 points. Studies that received lower MERSQI points were single pre–post studies, included only one institution, failed to report on the internal structure of the validity of the evaluation instrument, and did not report the response rate. Generally, studies had appropriate analysis, but due to their study design (single group pre-post studies), there were small sample sizes, and therefore, the complexity of the analysis did not exceed beyond a descriptive analysis.

### 3.3. Interventions

A summary of analyzed interventions is presented in Table 2.

Of 18 analyzed studies, in 3 studies, MBSR was used as an intervention, and its duration was 8 weeks in all cases. In eight studies, researchers used alternative, non-MBSR versions of mindfulness training. This was the most heterogeneous group among all interventions regarding duration and contents. In the remaining five studies, researchers used the following interventions: discussion groups [34,47], stress management and resilience training [37], the “Art of Seeing” course [42], and a wellness curriculum [43]. In two cases, mobile application was used as a source of mindfulness training [39,40].

### 3.4. Participants

In reviewed studies, 10 were dealing with physicians of different specialties (primary care, internal medicine, radiology, mixed/not specified;(Appendix A Appendix A), while in 8, residents/interns were also study subjects (Appendix A Appendix A).

### 3.5. Subgroup Analyses

Most commonly used psychometric instruments were Maslach Burnout Inventory (MBI) for burnout—used in eight studies; Perceived Stress Scale (PSS) for stress—administered in seven studies; Jefferson Scale for physician Empathy (JSE) for empathy—used in four studies; Five Facet Mindfulness Questionnaire (FFMQ) for mindfulness—used in three studies.

Mindfulness was assessed with the following five different instruments: Five Facet Mindfulness Questionnaire (FFMQ), Mindful Attention Awareness Scale (MAAS), Freiburg Mindfulness Inventory (FMI), Kentucky Inventory of Mindfulness Skills, and Two Factor Mindfulness Scale.

Maslach Burnout Inventory (MBI) was administered in eight studies. Within a subgroup of physicians, it was used in five studies [33,34,35,36,48]. In all studies, the result was a significant or substantial improvement in one or more burnout subscale scores. In all but one study [35], there was also sustained effect confirmed, and the duration of its measurement varied from 3 [48] to 12 months [34]. In the subgroup of residents, MBI was administered in three studies [39,41,43]. Taylor et al. [39] did not report exact MBI scores; results were presented only graphically. Bentley et al. [41] reported non-significant improvement in two subscales (EE and D) and a decrease in PA. Runyan et al. [43] reported positive trends in all measures of burnout post intervention, though they used different terminology: apart from the EE subscale, they used the “cynicism” and “professional efficacy” subscales.

### 3.6. The Duration of Interventions Effect

Another important question to address was how long the effects of specific intervention lasted. In 10 studies [33,34,35,36,37,43,44,46,47,48], the sustained effect was examined; the duration of measurement was from 2 to 15 months, with an average of 6.6 months. The results of these assessments generally suggest sustained or even augmented benefits for many months after completion of the studied intervention. One randomized controlled study [44] confirmed significant improvement in mindfulness and reduction in heart rate. Another RCT [34] confirmed improvement in elements of physician well-being, including meaning, empowerment, and engagement in work.

### 3.7. Beneficial Effects of Peer Support/Group Participation

There were two studies with qualitative design included [38,47]. In the first [38], authors conducted semi-structured interviews with primary care physicians who completed 52 h of a mindful communication program. The main topics reported from participants were as follows: (i) mindfulness skills improved the participant’s ability to be attentive and listen deeply to patients’ concerns and respond more effectively; (ii) sharing personal experiences from medical practice with colleagues reduced professional isolation; (iii) developing greater self-awareness was positive and transformative. The second study [47] was conducted in a hospital environment, and workplace-based discussion groups were conducted. The authors concluded that this intervention could provide a focus for an approach to professionalism that is mindful of self, the team, the culture, and the organization. Similar to a previous study, they agreed that sharing narratives of successes and challenges in professionalism could be a valuable experience [47].

## 4. Discussion

The aim of this systematic review was to analyze published literature about psychological interventions with elements of mindfulness (PIMs), considering duration, setting, and mode of administration. Regardless of the specific type of PIMs used, results generally demonstrated a positive impact on empathy, well-being, and reduction in burnout. The most commonly studied interventions involved MBSR, other mindfulness training programs, and discussion groups; all these strategies were proven effective in burnout prevention and reduction.

### 4.1. Psychological Interventions with Elements of Mindfulness Reducing Burnout in Physicians

In eight studies that reported on burnout using MBI [33,34,35,36,39,41,43,48], there was a significant improvement in at least one burnout subscale confirmed within the subgroup of physicians, whereas for residents there was only a positive statistically insignificant trend noticed. The difference between the two subgroups can be partially explained by the difference in their professional role—while physicians have more responsibility, residents face greater workload regarding patient’s counseling and treatment, which could have affected self-rated burnout scores. One of the included interventions for residents was based on a smartphone application as a 10 day program [39]; for the same intervention, MBI scores were not reported. Therefore, it could not have equal importance as other studies evaluating the impact on participants’ burnout. In the other two studies with residents as study subjects [41,43], there was a small sample (7 psychiatry residents and 12 family medicine residents), based on which generalizations could not be made.

### 4.2. The Sustained Effect of Psychological Interventions with Elements of Mindfulness

There were 10 studies that examined sustained effect (lasting 2 [35] to 15 [36] months, with an average of 6.6 months) of specific interventions, and the effect was generally confirmed. This is an intriguing starting point for possible further research, as the sustainability of measured changes is important information for physicians and their organizations, based on which they could plan possible capacity-building interventions to sustain the benefits.

When compared with other recently published systematic reviews [17,18,19,28,50,51], this paper is unique, as, to our knowledge, it is the only review with PIMs as an intervention, which is an added value, compared with those solely focusing on the duration of MBSR’s effects.

### 4.3. Psychological Interventions with Elements of Mindfulness as an Intervention Suitable for Physicians

Studied populations were comparable to other reviews, as interventions were also provided for physicians [17,18,28], resident physicians [19], general practitioners [51], and physicians and nurses [50]. Regardless of the differences, the main findings of this paper were synchronous with other systematic reviews—namely, that individual-focused and organizational strategies can result in clinically meaningful reductions in burnout among physicians [17,18,28,50]. In the case of resident physician burnout, the main finding of Busireddy et al. was that work hour limits were associated with improvement in emotional exhaustion and burnout [19].

In this review, there was only one study [32] designed to examine the influence of work hour limitation on stress and burnout in resident physicians; compared with a group using a 10-week mindfulness training program, in which significant stress and burnout reduction were observed, there were no such reductions for participants in the control group with working hour limitation identified.

Regarding the feasibility of interventions, we analyzed one study that used a combination of in-person and online training [35], and two studies [39,40] that applied modern technologies based interventions (mobile apps). In the first, Pflugeisen et al. [35] reported that eight-week video-module-based mindfulness intervention resulted in a significant decrease in stress and EE, an increase in mindfulness skills, and PA. In the second study [40], the four-week use of self-guided smartphone-based mindfulness application Headspace was observed, and there was a significant improvement in both positive affect and mindfulness scores but no change in negative affect scores.

These findings are important in light of an assumption that physicians may be less willing to commit to time-consuming, in-person training, compared with convenient, feasible, and less time-consuming online interventions. In line with this assumption, Wolever et al. conducted a randomized controlled trial with 239 participants using 2 delivery venues (online vs. in person) for the mindfulness intervention and reported that both produced equivalent results of the selected measures [52].

The feasibility and convenience are especially important in the present COVID-19 pandemic, which has placed a significant physical and psychological burden on physicians; therefore, protection and promotion of physicians’ well-being seem more crucial than ever [29,53].

In the group of general practitioners, researchers suggest an urgent need for high-quality, controlled studies on GPs’ well-being, as their health is threatened by the lack of (human) resources and workload issues [51]. The proportion of GPs intending to quit direct patient contact in the next five years continues to increase annually, with 60.9% of GPs over 50 years of age reporting this intention in a recent UK survey [51]. Within our review, there were two studies with GPs as study subjects [45,48]: Verweij et al. [45] reported fewer post-intervention burnout symptoms and increased work engagement and well-being but no increase in empathy, while Hamilton-West [48] observed significant improvement in validated measures of stress and burnout following the intervention, with maintained improvement at the three-month follow-up. However, their limitation is that both were pilot studies with a small sample size.

There is an intriguing correlation between increased mindfulness and improved patient care outlined by Krasner et al. [36]. Identified positive association between mindfulness and greater resiliency, and patient-centeredness suggests that mindfulness could have been an active component in promoting such changes [36].

### 4.4. Strengths and Limitations of the Review

The strength of this review is the adherence to the PRISMA reporting guidelines and the broad and clearly demonstrated nature of the search strategy. It is worth mentioning that a high number of studies were screened, and the assessment of methodological quality was performed using a well-regarded quality assessment tool.

Given that various instruments were administered for psychological qualities assessment, weak experimental designs (e.g., pre–post test designs) were applied and different durations of intervention and sustained effect measurement were analyzed, a comparison of results and generalization of main findings was challenging. Some studies used small samples and findings cannot be generalized to the population; in addition, some of the reports had a substantial risk of bias, mainly due to low ability to control the potential confounding factors. In other cases, there was a control group did not exist. Participants were mostly self-selected based on their level of interest, which could be another source of bias.

All included studies were focused solely on self-reported outcome measures, such as burnout, perceived stress, empathy, and well-being. Exclusive reliance on participant self-reports is limiting, as they were exposed to social desirability and acquiescence response biases. Therefore, future research is advised to upgrade self-reported data with objective physiological measures of stress and well-being or to combine subjective and objective data in order to validate the existing knowledge.

There were insufficient descriptions of the implemented interventions in several studies. Without comprehensive description and explication, interventions cannot be reliably replicated or implemented. For that purpose, guidelines were developed to improve reporting of intervention description and replication [54].

Based on the aforementioned limitations, a strong suggestion for future research is to be well planned, methodologically rigorous, and uniform in design, e.g., the same duration of intervention and sustained effect measurement, same psychometric measures for one quality arisen. This would contribute to advancing our knowledge of burnout in physicians, as well as the role of mindfulness-based techniques in addressing this problem.

## 5. Conclusions

Results of this systematic review demonstrate the positive impact of psychological interventions with elements of mindfulness (PIMs) on empathy, well-being, and reduction in burnout in physicians. The main finding is synchronous with other systematic reviews that individual-focused and organizational strategies can result in clinically meaningful reductions in physicians’ burnout, which is to be considered as a convincing and reliable outcome.

According to presented studies, it remains unclear which intervention offers the greatest value to physicians and their organizations. Future randomized controlled studies with a greater number of participants should address the optimal approaches toward the development and implementation of burnout reduction strategies, along with the assessment of the feasibility and associated costs.

One of the main questions yet to be addressed is the long-term impact and cost-effectiveness of different types, formats (online vs. in person), and dosages of specific PIMs for physicians. The sustained effects have remained unclear; therefore, it would be of utmost importance for psychological interventions with elements of mindfulness implementation to consider periodic re-exposure and frequency of refreshment courses to sustain or augment their effects.

This review provided sufficiently solid proof that it is possible to change attitudes toward care and improve several key aspects of burnout, well-being, and empathy solely by offering physicians effective interventions/programs focused on their needs and well-being, which should be a positive challenge for health care system management and physicians themselves during the COVID-19 pandemic and beyond.

Since the majority of interventions under review offered some kind of mindfulness training and/or discussion groups, which all were shown to be effective, we believe that the main contribution of this review is the “healing power” of peer support/group participation and mindfulness training. The latter is to be introduced in resident/trainee programs and organized as a means of psychosocial support and awareness-raising tool for physicians.

## Figures and Tables

**Figure 1 ijerph-18-11181-f001:**
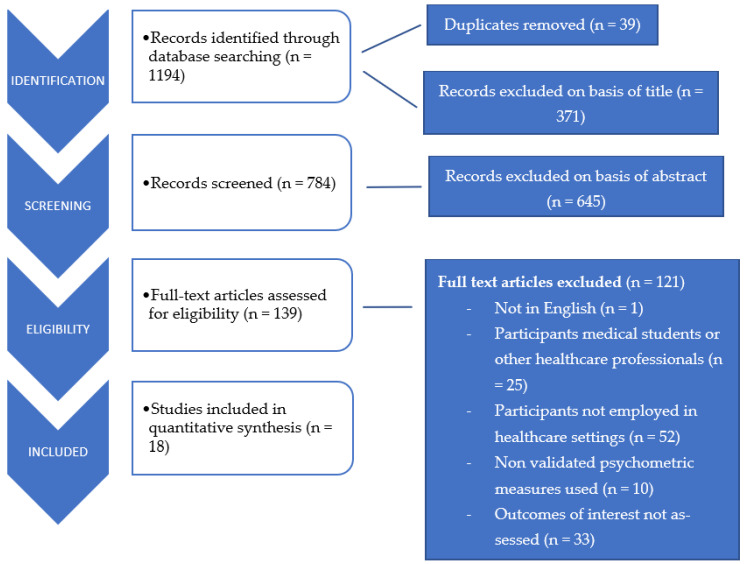
Study selection process.

**Table 1 ijerph-18-11181-t001:** Characteristics of reviewed studies.

Characteristics	Studies (*n*)
Study Location	
Australia [32]	1
USA [33,34,35,36,37,38,39,40,41,42,43]	11
Spain [44]	1
The Netherlands [45,46]	2
United Kingdom [47,48,49]	3
Participants	
Physicians [33,34,35,36,37,38,44,45,47,48]	10
Residents/interns [32,39,40,41,42,43,46,49]	8
Physician Specialty	
Primary care/Family medicine/General practice [33,36,38,45,48]	5
Internal medicine [34]	1
Radiology [37]	1
Not specified/mixed [35,44,47]	3
Intern doctors [32,49]	2
Residency mixed [40,42,46]	3
Pediatric residents [39]	1
Psychiatry residents [41]	1
Family medicine residents [43]	1
Methodological Design	
Randomized controlled trial [32,34,44]	3
Randomized controlled pilot study [37]	1
Prospective non-randomized pre–post intervention study [33,35,36,39,40,42,46,49]	8
Prospective non-randomized pre–post controlled intervention study [43]	1
Mixed method pre–post intervention study [41,45,48]	3
Qualitative study [38,47]	2

**Table 2 ijerph-18-11181-t002:** Summary of intervention characteristics.

Intervention Type	Studies; *n* (%)	Duration	Description
Mindfulness-based stress reduction (MBSR) [44,45,46]	3 (16.7)	8 weeks	A systematic approach to group mindfulness training offered on a weekly basis for eight consecutive weeks, usually with a silent retreat; classes delivered by a certified instructor.
Mindfulness training [32,33,35,36,38,41,48,49]	8 (44.4)	18 h–10 weeks	Various intervention designs based on mindfulness as a strategy for increasing concentration, awareness, and emotional regulation; using mindful meditation, movement, speaking, listening, compassion for self, and other mindfulness practices.
Discussion groups [34,47]	2 (11.1)	6–9 months	Elements of mindfulness, reflection, shared experience, professionalism, and small group learning.
Stress management and resilience training (SMART) [37]	1 (5.6)	90 min	Intentional awareness, attention, and attitude regulation practices, enabling participants to focus their attention on the external world and defer unrefined judgments.
“Art of Seeing” course [42]	1 (5.6)	4 weeks	Introduction to formal art analysis, introduction to symbols in art, mindful movement workshop, covering material beyond an introduction to conceptual and contemporary art.
Wellness curriculum [43]	1 (5.6)	1 month	Elements of mindfulness practice and breathing, meditation, narrative medicine, and cultivating gratitude.
Mindfulness intervention using smartphone application [39,40]	2 (11.1)	10 days/4 weeks	Using smartphone applications for a self-guided program based on mindfulness and meditation.

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
