# Peer review of "The Impact of Psychological Interventions with Elements of Mindfulness (PIM) on Empathy, Well-Being, and Reduction of Burnout in Physicians: A Systematic Review"

_ijerph, 2021, doi:10.3390/ijerph182111181_

Round 1

Reviewer 1 Report

It is a fine article with an inspiring vision of the burnout issue with a very specific population. The findings are of great significance, especially at the current moment.

Abstract:

A structural abstract may be better to conclude the information logically.

The suggestion "Future research should upgrade self-reported data with objective psychological measures and address the question, which intervention offers more benefits to healthcare professionals." may be too general and causal, please replace it with more precision.

Introduction:

Many latest pieces of research have been published on the issue, please provide more latest background to stress the importance of the study.

The structure of the whole introduction seems to be chaotic, please redesign it.

The authors mentioned COVID 19 in the abstract part but missed it anywhere else in the article. Please think about it and decide whether to mention it or not. If so, more related details need to be added.

Methods:

More details must be provided (e.g. date) to fully describe the methodology.

The inclusion & exclusion criteria seem to be indistinct, terms need to be clearer.

Results:

Tables and figures can be modified to exclude superfluous information.

The description of Table 2 is too general and lack citations to support it (whether it is from the authority or other institution).

Conclusion:

What is the practical meaning of the study? Not just for now, but for future use?

Is the result part sufficient for the conclusion? Convincing reasons need to be provided.

Please explain how the purpose of the study is realized.

Author Response

Response to Reviewer #1`s Comments

Abstract:

  1. R1: A structural abstract may be better to conclude the information logically.

Answer:

It was corrected. Please see lines 22-35 and further:

Introduction: Physician’s burnout has been recognized as an increasing and significant work-related syndrome, described by the combination of emotional exhaustion (EE) and depersonalization (D), together with low personal accomplishment (PA). It has many negative consequences on a personal, organizational and patient care level. This systematic review aimed to analyze research articles where psychological interventions with elements of mindfulness (PIMs) were used to support physicians in order to reduce burnout and foster empathy and wellbeing.

Methods: Systematic searches were conducted in May 2019, within six electronic databases PubMed, EBSCOHost Medline, PsycARTICLES, Cochrane Library, JSTOR and Slovenian national library information system. Different combinations of Boolean operators were used - mindfulness, empathy, medicine/family medicine/general practice/primary care, burnout, doctors/physicians, intervention and support group. Additional articles were hand-searched from the reference list of the included articles. Studies with other healthcare professionals (not physicians and residents) and/or medical students, and those where PIMs were applied for educational or patient’s treatment purposes were excluded.

  1. R1: The suggestion "Future research should upgrade self-reported data with objective psychological measures and address the question, which intervention offers more benefits to healthcare professionals." may be too general and causal, please replace it with more precision.

Answer:

We narrowed target group of healthcare professionals to physicians (line 46), in order to gain more homogeneous group and consistency for deriving conclusions from review. The suggestion of upgrading self-reported data with objective measures is described more precisely in section 4.4. Strengths and limitations of the review, lines 344 to 349.

Introduction:

  1. R1: Many latest pieces of research have been published on the issue, please provide more latest background to stress the importance of the study.

Answer: 

The structure of introduction is redesigned in a more consistent way, there is also some later research added (line 76 to 80) and references updated (ref.no 2,4,5,6,13,26,27,28,29,30).

  1. R1: The structure of the whole introduction seems to be chaotic, please redesign it.

Answer: 

The structure of introduction is redesigned with two subsections added: 1.1. Mindfulness and psychological interventions with elements of mindfulness and 1.2. Aim and research questions emphasized.

  1. R1: The authors mentioned COVID 19 in the abstract part but missed it anywhere else in the article. Please think about it and decide whether to mention it or not. If so, more related details need to be added.

Answer:

Authors agreed that review is relevant in the time of COVID 19 pandemic, as burdened physicians are enabled with an overview of effective and feasible interventions to improve their wellbeing, therefore related details on this topic were added (lines 56-59, 101-107).

Methods:

  1. R1: More details must be provided (e.g. date) to fully describe the methodology.

Answer:

Date of systematic searches is added. Inclusion and exclusion criteria are redesigned and clearly presented (lines 154-162). Main adoption is that another exclusion criterion is added, therefore studies focused on other healthcare professionals (not physicians and/or residents) and medical students were excluded. Study selection is supported by Figure 1, for more clarity.

Please see lines 131-148:

Detailed search strategy is outlined in Figure 1 and described as follows:

On PUBMED there were 246 papers in total from various searches: mindfulness AND intervention AND doctors (n=213); mindfulness AND doctors AND empathy (n=33). Given the search limits, papers were published in last ten years (2009-2019), written in English and concerned human beings.

On EBSCOHost, 247 papers were found using key words doctors AND mindfulness (n=19); mindfulness AND doctors AND empathy (n=228) with search limitation to publications in last five years.

In PsycARTICLES, a total of 417 research papers published in last ten years using key words support group AND medicine AND empathy (n=263); mindfulness AND medicine AND empathy (n=73); mindfulness AND family medicine (n=81) were extracted.

In Cochrane Library, there were 96 systematic reviews found for mindfulness (intervention), while in JSTOR, a total of 154 publications for last ten years using key words mindfulness AND doctors AND intervention (n=42); mindfulness AND empathy AND doctors (n=21); mindfulness AND medicine AND empathy (n=61); mindfulness AND intervention AND medicine AND doctors (n=30) were identified, and in COBISS, 34 papers were found using different search terms (mindfulness, medicine (n=16); mindfulness, burnout (n=8); mindfulness, empathy (n=10).

  1. R1: The inclusion & exclusion criteria seem to be indistinct, terms need to be clearer.

Answer:

Please see lines 154-162 – text was rewritten:

Research studies with quantitative, qualitative and mixed-method design (i), written in English or Slovenian language (ii); published in last ten years (or five years for EBSCOHost only (iii)) and presenting original research focused on evaluation of psychological interventions with elements of mindfulness (PIMs), implemented with the intention of burnout reduction and wellbeing promotion in physicians, including residents (iv) were included in this review.

Studies with psychological interventions used for patients` treatment or educational purposes (i) and those focused on other healthcare professionals (not physicians and/or residents) and medical students (ii), were also excluded.

Results:

  1. R1: Tables and figures can be modified to exclude superfluous information.

The description of Table 2 is too general and lack citations to support it (whether it is from the authority or other institution).

Answer

Table 1 and Table 2 are modified in accordance with updated study selection and to extract more relevant information. In Table 1 we excluded last characteristic “Intervention”, as intervention characteristics are presented separately in Table 2. In both tables (Table 1 and Table 2) there are also citations added.

Conclusion:

  1. R1: What is the practical meaning of the study? Not just for now, but for future use?

Answer:

Please see lines 377-381 as we believe that the practical meaning is explained in the text below:

This review provided a solid enough proof that it is possible to change attitudes towards care and improve several key aspects of burnout, well-being and empathy, solely by offering physicians effective interventions/programs focused on their needs and well-being, which should be a positive challenge for health care system managements and physicians themselves during the COVID-19 pandemic and later on.

And also lines 382-387:

Since the majority of interventions under review offered some kind of mindfulness training and/or discussion groups, which all were shown to be effective, we believe that main contribution of this review is the “healing power” of the peer-support/group participation and mindfulness training. The latter is to be introduced in resident/trainee programs and organized as a mean of psychosocial support and awareness raising tool for physicians.

  1. R1: Is the result part sufficient for the conclusion? Convincing reasons need to be provided.

Answer:

Please see lines 360-365 – reasons for our conclusions are provided below:

Results of this systematic review demonstrate positive impact of psychological interventions with elements of mindfulness (PIMs) on empathy, wellbeing and reduction of burnout in physicians. The main finding is synchronous with other systematic reviews that individual-focused and organizational strategies can result in clinically meaningful reductions in physicians’ burnout, which is to be considered as a convincing and reliable outcome.

  1. R1: Please explain how the purpose of the study is realized.

Answer:

Please see 254-260, at the beginning of the Discussion section – the realized purpose is explained in the below text:

The aim of this systematic review was to analyze published literature about psychological interventions with elements of mindfulness (PIMs), considering duration, setting and mode of administration and shown that regardless the specific type of PIMs, results generally demonstrated positive impact on empathy, wellbeing and reduction of burnout. The most commonly studied interventions involved MBSR, other mindfulness trainings and discussion groups, and all these strategies were proven as effective in burnout prevention and reduction.

The Response to Reviewer 1 is also attached.

Reviewer 2 Report

You can find attached a document with the suggestions.

Author Response

Response to Reviewer #2`s Comments

Title

  1. R2: I would recommend the authors to reduce the title, I think it would be more clear for the readers. Maybe you could use the term “healthcare professional” as you present in line 66.

Answer:

Title was modified in concordance with modified inclusion and exclusion criteria, by which subgroups of medical students and healthcare professionals are excluded in order to gain more homogeneous and relevant groups for conclusions, therefore we use the term “physicians” only.

Abstract

  1. R2: Burnout definition is important and it should appear in the abstract, and some methodology aspects as well.

Answer:

The mentioned two aspects are added in the abstract – please see lines 22-35:

Introduction: Physician’s burnout has been recognized as an increasing and significant work-related syndrome, described by the combination of emotional exhaustion (EE) and depersonalization (D), together with low personal accomplishment (PA). It has many negative consequences on a personal, organizational and patient care level. This systematic review aimed to analyze research articles where psychological interventions with elements of mindfulness (PIMs) were used to support physicians in order to reduce burnout and foster empathy and wellbeing.

Methods: Systematic searches were conducted in May 2019, within six electronic databases PubMed, EBSCOHost Medline, PsycARTICLES, Cochrane Library, JSTOR and Slovenian national library information system. Different combinations of Boolean operators were used - mindfulness, empathy, medicine/family medicine/general practice/primary care, burnout, doctors/physicians, intervention and support group. Additional articles were hand-searched from the reference list of the included articles. Studies with other healthcare professionals (not physicians and residents) and/or medical students, and those where PIMs were applied for educational or patient’s treatment purposes were excluded.

Introduction

  1. R2: Why is this review important? You should justify your research in the introduction.

Answer:

Introduction is redesigned and refreshed by more update references and relevant information to support the research.

Please see lines 95-112 – justification was added in the Introduction section:

Main advantage of the review is that it offers thorough insight into psychological interventions for physicians, which could help to reduce burnout. The topic is of utmost importance, as the COVID-19 pandemic heightened existing challenges for physicians, e.g. increasing workload, which was shown to be directly correlated with increased burnout [29].

Another unique characteristic is that in-person and modern information technologies based interventions (mobile apps) were analyzed. Given that studies showed online and other applications/interventions effectively ameliorate burnout and alleviate anxiety and depression in physicians and medical residents, these findings have an additional value in the context of COVID-19 reality. Therefore, burdened physicians are enabled with effective and feasible interventions to help them improve their wellbeing. Moreover, online interventions are easily accessible, less time consuming and user friendly [30].

It also stresses the importance of sustained effect of the selected PIMs. Regarding type of intervention, this review is a unique achievement in comparison with others, since all psychological interventions with elements of mindfulness (PIMs) were included and provided ground for broader understanding of PIMs, which is an additional value, compared to a sole demonstration of the MBSR effects.

  1. R2: When talking about PIM, is it plural or singular? Because it could be understood as a plural noun but in line 86 you use the singular form of the verb.

Answer:

PIMs are plural, the abbreviation was corrected throughout the text.

Methods

  1. R2: Describing the search strategy in a table would be more clear and visual than doing it in the text

Answer:

Please see lines 131-148:

Detailed search strategy is outlined in Figure 1 and described as follows:

On PUBMED there were 246 papers in total from various searches: mindfulness AND intervention AND doctors (n=213); mindfulness AND doctors AND empathy (n=33). Given the search limits, papers were published in last ten years (2009-2019), written in English and concerned human beings.

On EBSCOHost, 247 papers were found using key words doctors AND mindfulness (n=19); mindfulness AND doctors AND empathy (n=228) with search limitation to publications in last five years.

In PsycARTICLES, a total of 417 research papers published in last ten years using key words support group AND medicine AND empathy (n=263); mindfulness AND medicine AND empathy (n=73); mindfulness AND family medicine (n=81) were extracted.

In Cochrane Library, there were 96 systematic reviews found for mindfulness (intervention), while in JSTOR, a total of 154 publications for last ten years using key words mindfulness AND doctors AND intervention (n=42); mindfulness AND empathy AND doctors (n=21); mindfulness AND medicine AND empathy (n=61); mindfulness AND intervention AND medicine AND doctors (n=30) were identified, and in COBISS, 34 papers were found using different search terms (mindfulness, medicine (n=16); mindfulness, burnout (n=8); mindfulness, empathy (n=10).

Please see Figure1:

  1. R2: This affects also the inclusion criteria, the paragraph is too long and it is not clear for the reader.

Answer:

Please see lines 154-162:

Research studies with quantitative, qualitative and mixed-method design (i), written in English or Slovenian language (ii); published in last ten years (or five years for EBSCOHost only (iii)) and presenting original research focused on evaluation of psychological interventions with elements of mindfulness (PIMs), implemented with the intention of burnout reduction and wellbeing promotion in physicians, including residents (iv) were included in this review.

Studies with psychological interventions used for patients` treatment or educational purposes (i) and those focused on other healthcare professionals (not physicians and/or residents) and medical students (ii), were also excluded.

  1. R2: In line 138 you don’t explain what happened in case of disagreement between the two reviewers. You usually need a third reviewer, did you use it?

Answer:

Search strategy of study selection is refreshed and clearly presented in Figure 1, with details described in text (line 124 to 130).

Paragraph with inclusion and exclusion criteria is corrected, and another exclusion criterion is added.

In the process of data extraction we used a third reviewer, all are specified in lines 164 and 165.

Results

  1. R2: 43 studies is a huge number for a review. Have you thought of adding other inclusion

criteria in order to reduce this number?

Answer:

Authors are grateful for this suggestion regarding inclusion criteria, which we followed. After a thorough discussion we added another exclusion criterion where other healthcare professionals (not physicians and/or residents) and medical students were excluded. By doing that, number of studies was reduced to 18 and with more homogeneous group of physicians results could be more efficiently discussed and conclusions made.

Discussion

  1. R2: The sentence from line 289 to line 291 could be expressed in a more clear way, it is not easy to understand.

I recommend the authors to improve the structure and expression of the discussion, it could be redacted in a more clear way.

Answer:

The whole discussion is redesigned, we believe in a more clear way. Four subsections were made to make Discussion section more clear:

4.1. Psychological interventions with elements of mindfulness reducing burnout in physicians

4.2. The sustained effect of Psychological interventions with elements of mindfulness

4.3. Psychological interventions with elements of mindfulness as an intervention suitablje for physicians

4.4. Strengths and limitations of the review

Discussion and Conclusions sections were rewritten and improved.

The Response to Reviewer 2 is also attached.

Round 2

Reviewer 2 Report

I would like to congratulate the authors for their hard work improving the manuscript, it is much more clear and structured now. The reduced number of final manuscripts help to achieve better conclusions as well...

I don't have further comments on your manuscript, best regards